# Microtubule Organization in Striated Muscle Cells

**DOI:** 10.3390/cells9061395

**Published:** 2020-06-03

**Authors:** Robert Becker, Marina Leone, Felix B. Engel

**Affiliations:** 1Experimental Renal and Cardiovascular Research, Department of Nephropathology, Institute of Pathology, Friedrich-Alexander-Universität Erlangen-Nürnberg (FAU), 91054 Erlangen, Germany; robert.becker@uk-erlangen.de (R.B.); marina.leone@i-med.ac.at (M.L.); 2Division of Developmental Immunology, Biocenter, Medical University of Innsbruck, 6020 Innsbruck, Austria; 3Muscle Research Center Erlangen (MURCE), 91054 Erlangen, Germany

**Keywords:** centrosome, MTOC, non-centrosomal MTOC, skeletal muscle, cardiomyocytes, cell cycle, microtubules

## Abstract

Distinctly organized microtubule networks contribute to the function of differentiated cell types such as neurons, epithelial cells, skeletal myotubes, and cardiomyocytes. In striated (i.e., skeletal and cardiac) muscle cells, the nuclear envelope acts as the dominant microtubule-organizing center (MTOC) and the function of the centrosome—the canonical MTOC of mammalian cells—is attenuated, a common feature of differentiated cell types. We summarize the mechanisms known to underlie MTOC formation at the nuclear envelope, discuss the significance of the nuclear envelope MTOC for muscle function and cell cycle progression, and outline potential mechanisms of centrosome attenuation.

## 1. Introduction: Non-Centrosomal Microtubule-Organizing Centers—A Hallmark of Differentiation

Microtubules are an integral part of the cytoskeleton, playing important roles in cellular processes such as intracellular trafficking, cell division, and maintenance of cellular architecture including shape, polarity, and organelle positioning. In proliferating animal cells, the majority of microtubules are organized by an organelle termed the centrosome, which is therefore labeled the dominant microtubule-organizing center (MTOC). One of the best-known roles of the centrosomal MTOC is to ensure the proper formation and positioning of the bipolar spindle during cell division [1]. As cells differentiate to perform organ- or tissue-specific functions, their microtubule network re-organizes to meet the new functional demands. In the course of re-organization, the centrosomal MTOC function is attenuated and microtubules are mainly organized from other subcellular sites, consequently termed non-centrosomal MTOCs (ncMTOCs) [2]. This shift from a centrosomal MTOC to ncMTOCs occurs to varying degrees in different cell types. Keratinocytes for example continue to generate microtubules from the centrosome but, subsequently, centrosome-generated microtubules are distributed and anchored to the apical membrane [3]. By contrast, neurons and striated (i.e., skeletal and cardiac) muscle cells only retain vestigial MTOC activity at centrosomes while dominant MTOC function is exhibited by non-centrosomal sites in these cells [4,5,6].

Transition from centrosomal MTOCs to ncMTOCs occurs mainly in cells that—under physiological conditions—permanently exit the cell cycle during differentiation. Prominent examples are differentiated epithelial cells, cardiomyocytes, skeletal myotubes, and neurons [4,7,8,9,10,11,12,13]. Considering the role of the centrosome in cell division, it appears logical that cells, which attenuate their centrosomal MTOC, also withdraw from a proliferative state. On the other hand, cell cycle exit could be a pre-requisite which allows cells to reduce the dominance of the centrosomal MTOC in order to establish ncMTOCs. How the relationship between cell cycle status and MTOC regulation contributes to proper function of terminally differentiated cells remains elusive for most cell types. In this review, we focus on striated muscle (1) to summarize mechanisms of ncMTOC formation, (2) to illuminate diverse functional aspects of ncMTOCs, and (3) to discuss the interplay of ncMTOC formation and cell cycle.

## 2. Microtubule Organization and Function

Microtubules are dynamic, hollow filaments made up of α- and β-tubulin heterodimers, which are organized in a lattice of protofilaments. In animal cells, the number of protofilaments in the lattice varies from 11 to 15, but canonically, microtubules consist of 13 protofilaments [14]. The asymmetry of the αβ-tubulin heterodimers and the specific head-to-tail fashion in which they align confer an intrinsic polarity to microtubules [15,16,17]. The microtubule plus end, where β-tubulin is exposed, grows and shrinks faster in vitro when compared to the α-tubulin-exposing minus end [18]. Microtubules grow by the addition of guanosine triphosphate (GTP)-bound αβ-tubulin heterodimers (i.e., GTP cap), whereas shrinkage occurs when the heterodimer lattice is destabilized due to hydrolysis of GTP to guanosine diphosphate (GDP) [18]. While microtubules can nucleate spontaneously from purified tubulin in vitro, this process is kinetically restrained and requires comparably high tubulin concentrations [19,20]. Cells have therefore developed various mechanisms to modulate the kinetics of microtubule assembly and to regulate their microtubule network in a dynamic fashion.

### 2.1. Control of Microtubule Dynamics

Controlling microtubule dynamics is pivotal for cells, as microtubules serve as tracks for intracellular transport and contribute to cellular organization and plasticity by providing scaffolds for and exerting forces on subcellular structures. While plus ends mainly contribute to microtubule mass and dynamic interactions of microtubules with other subcellular structures, nucleation and stable anchoring of minus ends determines microtubule network organization [21]. A key factor for microtubule nucleation is the γ-tubulin ring complex (γTuRC) which is formed from γ-tubulin and several γ-tubulin ring complex proteins (GCPs) [22]. The γTuRC acts as a template to kinetically favor the initial assembly of so-called “microtubule seeds” and stabilizes microtubules by capping their minus ends [22,23]. Consequently, a main feature of MTOCs is the concentration of proteins that have the ability to recruit γTuRC components and to promote γTuRC formation (see Section 3). Additional factors that, similar to γTuRC, control microtubule dynamics by capping and stabilizing minus ends are, among others, members of the calmodulin-regulated spectrin-associated protein (CAMSAP)/Patronin family (reviewed in [21]).

Another important group of proteins regulating microtubule dynamics are Tog domain-containing microtubule polymerases of which the microtubule-associated protein 215 kDa (XMAP215) is the most prominent example [24]. Tog domain proteins associate with γTuRCs or other microtubule templates and promote the longitudinal addition of new αβ-tubulin dimers [24,25]. The Ran pathway target TPX2 (targeting protein for Xklp2) is thought to act in synergy with XMAP215 by stabilizing early microtubule nucleation intermediates [22,26,27]. It is important to note that the mode of controlling microtubule dynamics likely varies between cell types and MTOCs [22]. While γTuRC-dependent microtubule assembly is kinetically dominant at most MTOCs [22,28], microtubules can also be nucleated in the absence of γTuRCs in vitro and in vivo [22]. It has therefore been proposed that the main function of γTuRCs, rather than allowing microtubule nucleation per se, is to define microtubule polarity and to ensure efficient and precise assembly of the 13 protofilament arrangement found in most microtubules [14].

Microtubule dynamics are also controlled by the incorporation of different tubulin isoforms [29] and posttranslational modifications (PTMs) of tubulin, collectively referred to as the “tubulin code” (reviewed in [16]). In humans, at least nine genes each exist for α- and β-tubulin [16,30,31]. A prominent example for the regulation of microtubule dynamics by tubulin isoforms is the β3 isoform predominantly expressed in neurons [32]. Microtubules assembled with β3-tubulin are less resistant to depolymerization [33,34] which has been suggested to underlie the more dynamic microtubule network in neurons [16,35]. Among tubulin PTMs, tyrosination and detyrosination, acetylation, and polyglutamylation are best studied [16]. These PTMs control microtubule dynamics mainly by regulating the interaction of microtubules with microtubule associated proteins (MAPs) [16]. Detyrosination stabilizes microtubules by preventing kinesin-13-mediated active depolymerization [36] and by reducing microtubule growth [37]. Acetylation stabilizes microtubules by protecting them from mechanical ageing and eventual breakage due to repetitive bending [38]. Polyglutamylation modulates severing of microtubules by spastin and katanin [39,40,41] and regulates the interaction of microtubules with a number of additional MAPs that potentially contribute to the control of microtubule dynamics [16,42,43]. In addition to microtubule properties, the tubulin code also contributes to the versatile roles of microtubules in the cell (reviewed in [16]).

### 2.2. Control of Microtubule Function

A central function of microtubules is to segregate chromosomes during mitosis. Chromosome segregation is accomplished by a bipolar spindle consisting of a variety of microtubules. Astral microtubules, emanating from the centrosome (see Section 3), contact the cell cortex with their plus-ends and are known to be fundamental for spindle orientation and elongation [44,45,46,47]. Kinetochore microtubules (or k-fibers) connecting the kinetochore of each sister chromatid to one spindle pole are required for faithful chromosome segregation [48]. Interpolar microtubules balance the forces generated in the spindle by kinetochore microtubules [49] and contribute to spindle bipolarity and central spindle assembly [47].

Besides chromosome segregation, microtubule organization is pivotal for a large number of cellular processes. This includes the regulation of cell shape by interacting with actin and actomyosin [50,51] and intracellular transport by acting as transport routes for a magnitude of cargo such as mRNA, protein and vesicles [52,53]. Furthermore, microtubules are involved in: (1) Formation of primary cilia [54] as well as motile cilia contributing to cell movement [55] or the generation of liquid flow [56]. (2) Positioning of organelles and intracellular organization [57,58,59,60,61,62,63,64,65]. (3) Mechanotransduction [66].

Many microtubule-dependent functions are exerted by microtubule-associated motors which are essential for microtubules to function as transport routes [67,68]. At the same time, motor proteins contribute to microtubule network organization by moving or sliding microtubules along each other [69,70]. Two main groups of microtubule-associated motors can be distinguished: the dynein complex, which moves in the direction of microtubule minus ends and kinesins, which primarily exhibit plus end-directed movement [71,72]. These opposing movements are complemented by the ability of dynein and kinesin to transport the respective other to opposing microtubule ends [73,74]. These two activities allow cells to control and fine tune bidirectional movement along microtubules. 

Dynein is present in two isoforms that mediate movement in the cytoplasm (dynein 1) and in cilia (dynein 2) [71,75]. Here, we will only summarize cytoplasmic dynein 1 functions, subsequently referred to as dynein. To be active, dynein forms a complex with dynactin [76]. Additionally, it interacts with activating adaptors like bicaudal D homologue 2 (BICD2) that link the motor complexes to various cargos like proteins, vesicles, or organelles (reviewed in [77]). Apart from moving along microtubules for transport purposes, dynein can be anchored at subcellular sites to exert pulling forces on other structures via microtubules [78,79,80]. One of the best studied examples is the anchoring of dynein by a trimeric complex consisting of the GDP-loaded Gαi subunit of heterotrimeric G-proteins, the adapter protein LGN (leucine–glycine–asparagine), and the dynein-binding nuclear mitotic apparatus protein (NuMA) to the mitotic cell cortex [81,82,83]. There, dynein exerts forces on astral microtubules to shape and position the bipolar spindle [84].

Kinesins are a large, heterogeneous superfamily, which is comprised of 14 families with 45 members in humans [85] that exert different microtubule-associated functions [72]. The kinesin superfamily regulates microtubule dynamics (e.g., members of the kinesin-8 [86] and kinesin-13 [87] families) and transport of intracellular cargo (e.g., kinesin-1 [88], kinesin-2, and kinesin-3 family members [89,90]). Transport and positioning of organelles are mediated by members of the kinesin-1 and kinesin-5 family [61,91].

## 3. MTOCs—the Centrosome and Beyond

As recently suggested by Joukov and De Nicolo, any structure that organizes microtubules by regulating their nucleation and/or anchoring should be termed an MTOC [92]. Given the pivotal role of microtubules for various aspects of cellular behavior, all types of eukaryotic cells have developed their own types of MTOCs to organize the microtubule cytoskeleton in a way that fits their needs best. The types of different MTOCs and their evolution have been excellently reviewed before [92,93,94,95]. In the following, we will focus on types of MTOCs in animal cells.

### 3.1. The Centrosome

In proliferating animal cells, the main MTOC is the centrosome (Figure 1), which was first described in 1887 independently by Edouard van Beneden and Theodor Boveri, who also coined the name to the organelle that he saw at “the centres of the forming daughter cells [after cell division], around which all other cellular components arrange themselves symmetrically” [96]. Of note, the only animal completely devoid of centrosomes known to date are planarians, a species of flatworms [97]. The centrosome is commonly defined as a juxtanuclear, non-membranous organelle consisting of two tubulin-based barrel-shaped centrioles at its core which are surrounded by a dense multiprotein cloud termed the pericentriolar material (PCM) [98]. The PCM contains various microtubule-organizing proteins and therefore accounts majorly for the MTOC activity of centrosomes [99]. While some PCM components can self-organize into foci, centrioles are considered to be essential for efficient PCM assembly [100,101]. Super-resolution microscopy indicated that the “basic” PCM in interphase cells (in contrast to the expanded PCM in mitotic cells; see below) is highly organized [99,102,103,104,105]. The PCM proteins pericentrin (PCNT) and centrosomal protein (CEP) of 152 kDa (CEP152) interact directly with the centriole wall and form radial fibers that extend away from the centrioles. In the perimeter delimited by these radial fibers, other PCM proteins like CEP192, CEP215 (also known as CDK5RAP2), and Polo-like kinase 1 (PLK1) are organized in layered domains. Ultimately, γTuRCs are recruited to the PCM and, in turn, promote microtubule nucleation at centrosomes [22,23]. The mechanisms of γTuRC recruitment vary between organisms and during the cell cycle. While several PCM proteins can directly interact with γTuRCs via an evolutionary conserved centrosomin motif 1 (CM1) [22], two recent studies indicate that CEP192, but not PCNT or CEP215, is required for the majority of centrosomal microtubule nucleation in interphase [106,107]. After being nucleated at the PCM, microtubules can be stably anchored at so-called subdistal appendages of centrioles by the PCM component ninein [108]. In interphase, ninein also contributes to γTuRC recruitment to the PCM [109]. Another key component of the centrosomal MTOC are centriolar satellites [110], which are cytoplasmic granules that move along microtubules mainly in direction of minus ends and therefore concentrate in the vicinity of the centrosome. Centriolar satellites contribute to centrosome assembly presumably through transport of centrosomal proteins. A key component of centriolar satellites is the oligomerizing protein pericentriolar material 1 (PCM1) [111,112,113,114]. When PCM1 is depleted, centriolar satellites are dispersed and centrosomal levels of PCNT and ninein are reduced [112].

In mitosis, the PCM expands into a mesh-like matrix in a process termed “centrosome maturation” that results in a significant increase in microtubule nucleation [115]. CEP192 plays a key role in this expansion by forming a complex with PLK1 and Aurora Kinase A (AurA) and recruiting them to the centrosome [116,117]. AurA accumulation at the centrosome leads to its activation by autophosphorylation and, subsequently, to phosphorylation and activation of PLK1 [116]. Active PLK1 phosphorylates CEP192 and other PCM proteins, which form a mitotic scaffold that underlies PCM expansion [98,116,118,119,120]. Additionally, phosphorylated CEP192 increases microtubule nucleation activity at mitotic centrosomes by providing docking sites for complexes of the phosphorylated γ-tubulin interactor NEDD1 (neural precursor cell expressed, developmentally down-regulated protein 1, also known as GCP-WD) and γTuRCs [116,121,122,123].

### 3.2. Centrosome-Independent Microtubule Organizing Pathways

Despite the importance and omnipresence of centrosomes in proliferating animal cells, several centrosome-independent microtubule organizing pathways exist that usually act in synergy with centrosomes but can also regulate microtubule networks in the absence of centrosomes (Figure 1). The augmin/HAUS (homologous to augmin subunits) pathway promotes microtubule nucleation through the recruitment of NEDD1 and γTuRCs to the sides of pre-existing microtubules [123,124,125,126]. During mitosis, augmin-mediated microtubule nucleation is responsible for the majority of kinetochore microtubules (see Section 2.2) [127]. A second pathway, mainly involved in mitotic spindle formation, is regulated by the small GTP-binding protein Ran [128]. Effectors of this pathway—collectively referred to as spindle assembly factors (SAFs) [129,130,131]—are sequestered by importins but are released upon RanGTP binding [128]. The cytoplasmic Ran GTPase activating protein converts RanGTP to RanGDP, while the chromatin-associated guanine nucleotide exchange factor RCC1 promotes exchange of GDP with GTP [131]. The spatial distribution of these two opposing activities results in a Ran gradient with high levels of RanGTP (and, consequently, SAF release) proximal to chromatin [132,133,134]. Well known SAFs are the microtubule stabilizing factor TPX2 [135], which promotes AurA-dependent NEDD1 phosphorylation and, thus, microtubule nucleation [136], and NuMa, which acts together with dynein to cluster microtubules by their minus ends [137,138]. Additional SAFs include, among others, Anillin, APC, MCRS1, and the kinesins Kif2a and Kif14 (reviewed in [131]). The Ran pathway is particularly important in oocytes, which are—under physiological conditions—the only dividing cell type that degrades centrosomes [139,140,141]. While dividing mouse oocytes assemble numerous acentrosomal MTOCs containing PCNT, γ-tubulin, NEDD1, and CEP192 [142], a prominent acentrosomal MTOC in human oocytes remains elusive [143,144]. It must be noted that this correlates with a more error-prone chromosome segregation in human oocytes compared to mouse oocytes [144,145]. Yet, in both cell types, a Ran gradient allows local SAF activation which promotes microtubule aster formation as well as clustering of microtubules and (in case of mouse oocytes) MTOCs into a bipolar spindle [131,135].

### 3.3. Non-Centrosomal MTOCs

Besides centrosome-independent microtubule organizing pathways, various cell types organize ncMTOCs in a process, which involves the recruitment of centrosomal proteins to distinct subcellular sites (Figure 1). Studies in centrosome-ablated cells and detailed tracking of microtubule nucleation demonstrated that the Golgi apparatus (Golgi) acts as an MTOC in various cell types [146,147,148,149,150]. Microtubules are pivotal for Golgi integrity and positioning and oriented microtubule arrays act in synergy with proper Golgi positioning to define a secretory axis [151]. While several PCM proteins, such as A-kinase anchoring protein 9 (AKAP9, also known as AKAP450, AKAP350, and CG-NAP), PCNT, and CEP215, localize to the Golgi [151], microtubule nucleation depends specifically on AKAP9, which recruits γTuRCs to the *cis*-side of the Golgi [152]. However, CEP215 and myomegalin localize to the Golgi in an AKAP9-dependent manner and might contribute to γTuRC recruitment and proper microtubule nucleation [153,154]. In addition, a complex of AKAP9 and myomegalin recruits CAMSAP2-stabilized microtubule minus ends to the Golgi [155].

Non-centrosomal microtubule arrays have been observed in a variety of epithelial cells, ranging from *Drosophila* ovarian follicle and tracheal cells to intestinal cells of *C. elegans* and mammals [7,156,157,158,159]. In mammalian intestinal epithelial cells, the spectraplakin MACF1/ACF7 localizes CAMSAP-bound microtubules to the apical membrane to establish apico-basal polarity [157]. The role of MACF1/ACF7, CAMSAPs, and their respective orthologues in other species appears to be conserved in various epithelial cell types [156,160]. In *Drosophila* tracheal cells, the microtubule-severing enzyme Spastin promotes the localization of microtubules and γTuRCs to the apical membrane, where the membrane protein Piopio facilitates their attachment [8]. Other subcellular structures that can become the main MTOC include mitochondria (spermatids in *Drosophila* [161]) and the nuclear envelope (*Drosophila* fat body cells [162] as well as striated muscle cells in *Drosophila* and mammals, see Section 4).

In addition, there are cell types in which the origin of microtubule nucleation is variable. In neurons, centrosomes are attenuated and have been found to be dispensable for proper microtubule organization in axons and dendrites, which is critical for neuron function (recently reviewed in [163]). A dominant ncMTOC has been elusive so far, but it appears that several aforementioned centrosome-independent pathways regulate neural microtubule organization (Figure 1). First, *Drosophila* plp (orthologue of PCNT and AKAP9) recruits γTuRCs to dendritic Golgi outposts but also to Golgi-independent dendritic branching points [148,164,165]. Secondly, the augmin pathway contributes to neural microtubule organization, especially polarity [166,167]. Third is minus end stabilization by members of the CAMSAP family that regulates microtubule stability in dendrites [168,169]. Finally, other sites of localized microtubule organization have been suggested, including synaptic boutons and undefined cytoplasmic sites [163]. Overall, neural microtubule organization appears to be controlled by several pathways which possibly accounts for the differential microtubule organization in axons, dendrites, and cell body of a single neuron. 

## 4. ncMTOC Formation at the Nuclear Envelope of Striated Muscle Cells

The importance of microtubule cytoskeleton reorganization for myogenesis has been under study for a long time [9,170,171,172]. In steady state, microtubules were found organized in arrays parallel to the longitudinal axis of muscle cells. In an effort to decipher underlying principles for the uniqueness of microtubule organization in striated muscle, pioneering studies more than 30 years ago revealed that, in human myotubes, centrosomes lose their typical juxtanuclear position and that an anti-centrosome autoimmune serum labels the nuclear envelope, while in the same cells labeling intensity at the centrosome declines [9]. Regrowth experiments showed that new microtubules emanate mainly from the nuclear envelope, quickly re-establishing the parallel arrays observed in steady state. Thus, the authors concluded that MTOC function is re-assigned to the nuclear envelope during myogenic differentiation, while the centrosomal MTOC is attenuated. Subsequent electron microscopy studies showed that also postnatal rat cardiomyocytes contain two subpopulations of microtubules: “(1) those adjacent to the nucleus (perinuclear), and (2) those distributed between the myofilament bundles (non-perinuclear)” [170]. In chicken, it was confirmed that the nuclear envelope MTOC is a feature of striated muscle by showing anti-PCM serum labeling and microtubule regrowth at the nuclear envelope of cardiomyocytes but not smooth muscle cells [173].

### 4.1. Anchoring of Centrosomal Proteins and Control of Microtubule Nucleation

Since the initial discovery that the nuclear envelope in striated muscle cells becomes the dominant MTOC (see above), improved protocols and technologies, the availability of antibodies specific to various centrosomal components, and the vast increase in knowledge regarding centrosome structure and function resulted in a more detailed characterization of the nuclear envelope MTOC, though most experiments have been performed solely with skeletal muscle cells. It was shown that MTOC proteins can be detected at the nuclear envelope of differentiating muscle cells prior to fusion into myotubes. Furthermore, microtubules can be observed in “sun-like” arrays around the nucleus in these differentiating mono-nucleated cells in contrast to the centrosome-originating star-like microtubule network in proliferating myoblasts [10]. Yet, whether microtubule nucleation already occurs at the nuclear envelope of differentiating mononuclear cells has not been assessed [10,174]. To date, the PCM proteins PCNT, ninein, CEP215 and AKAP9, the centriolar satellite component PCM1, and the γTuRC core element γ-tubulin have been found to localize to the nuclear envelope [4,10,60,175] (Figure 2). Notably, studies from different groups suggest that recruitment of these MTOC components during muscle differentiation occurs independently of microtubules or actin filaments [176,177]. Microinjection using inhibitory antibodies indicated that, similar to the situation at other MTOCs, γ-tubulin is pivotal for microtubule nucleation at the nuclear envelope [175]. Several centrosomal proteins have the ability to recruit γTuRCs, but depletion experiments indicated that, at the nuclear envelope, AKAP9 is specifically required for the nucleation of microtubules, whereas PCNT and CEP215 are dispensable [60]. While the ability of AKAP9 to recruit γTuRCs has been well described at the Golgi (see Section 3.3), it remains unclear why other γTuRC-interacting proteins do not contribute notably to microtubule nucleation at the nuclear envelope. Interestingly, a recent study in *Drosophila* showed that at the perinuclear ncMTOC of fat body cells, microtubule nucleation depends on the CAMSAP orthologue Patronin and the microtubule polymerase Msps (XMAP215 family member) but not on γ-tubulin [162]. Future studies should aim to dissect the contributions of γTuRCs and other microtubule polymerizing (e.g., XMAP215) or stabilizing (e.g., CAMSAPs, TPX2) factors to microtubule nucleation at the nuclear envelope of striated muscle cells.

Anchoring of MTOC proteins depends on a muscle-specific isoform of the outer nuclear membrane protein nesprin-1, nesprin-1α [60,178], which belongs to the Klarsicht, ANC-1, and Syne homology (KASH) protein family [179,180]. KASH proteins form, together with Sad1 and UNC-84 (SUN) proteins at the inner nuclear membrane, the linker of nucleoskeleton and cytoskeleton (LINC) complex [181,182,183]. Depletion of nesprin-1α dispersed PCM1, PCNT, and AKAP9 from the nuclear envelope and premature expression of nesprin-1α in undifferentiated (i.e., myogenin-negative) myoblasts was sufficient to recruit a fraction of endogenous PCM1 as well as an ectopically expressed centrosome-targeting domain of PCNT and AKAP9 (i.e., PACT domain) [60]. Yet, it remains unclear whether nesprin-1α is sufficient (1) to recruit endogenous PCM proteins in myoblasts, or (2) to induce nuclear envelope recruitment of MTOC components in non-muscle cells.

Apart from the anchor nesprin-1α not much is known about the mechanisms underlying the transition from centrosomal to non-centrosomal MTOC. Interestingly, nuclei of non-muscle cells recruit centrosomal proteins upon induced fusion with myotubes independently of new protein synthesis [177]. This suggests that mechanisms controlling the localization of centrosomal proteins to the nuclear envelope are constantly active.

PCNT is alternatively spliced in skeletal muscle and heart [184]. In cardiomyocytes, this splicing isoform of PCNT has been shown to preferentially localize to the nuclear envelope [4]. However, its mechanistic relevance for ncMTOC formation has yet to be determined. The involvement of alternative splicing in ncMTOC formation is underlined by a study in *Drosophila* spermatids. *Drosophila* centrosomin (Cnn; orthologue of CEP215) is spliced into a testes-specific isoform (CnnT) which contains a *C*-terminal mitochondrial-targeting domain [161]. CnnT recruits γTuRCs and converts mitochondria to ncMTOCs.

In *Drosophila* muscle, RacGAP50C and Pavarotti (orthologue of KIF23/MKLP1) have been implicated in non-centrosomal microtubule organization [185] (Figure 2). Both proteins are canonically known for their roles in spindle assembly and actomyosin ring formation during mitosis [186,187]. Furthermore, they have been shown to play a role in microtubule polarity and proper neural morphogenesis [188,189]. Similarly, RacGAP50C and Pavarotti are required for uniform microtubule polarity in fly muscle [185]. In addition, it has been suggested that they contribute to the perinuclear localization of γ-tubulin. Although conclusive evidence is missing, the hypothetical model is that γ-tubulin interacts with RacGAP50C and that Pavarotti, which localizes mainly to nuclei in muscle fibers, recruits RacGAP50C [185]. Whether microtubule motor activity of the kinesin Pavarotti is required for the recruitment of RacGAP50C or if Pavarotti merely acts as an adapter between the nuclear envelope and RacGAP50C is unclear. Furthermore, it needs to be elucidated, whether Pavarotti and RacGAP50C are involved in the initial MTOC formation at the nuclear envelope or if they only maintain perinuclear γ-tubulin localization in myofibers. More studies are required to decipher the role of RacGAP50C and Pavarotti in ncMTOC formation and to determine if a similar mechanism is conserved in mammals.

Importantly, while the nuclear envelope exhibits dominant MTOC function in differentiating muscle cells, the majority of microtubules are organized in longitudinal bundles parallel to the long axis of mature muscle cells [9,190]. In *Drosophila*, microtubules are anchored to the muscle cell poles via the cytoplasmic linker protein 190 (CLIP-190) [191]. Microtubules that have been nucleated at the nuclear envelope might be transported to the cell cortex, possibly by motor-dependent processes [192], and then anchored by CLIP-190. Besides being cortically anchored, it was found in mouse that microtubule bundles originate from node-like structures that co-localize with Golgi and endoplasmic reticulum (ER) exit site markers and, consequently, it was concluded that the Golgi serves as an ncMTOC in mature muscle cells [190]. Interestingly, AKAP9, which plays a main role for Golgi-derived microtubules (see Section 3.3), is also required for microtubule nucleation at the nuclear envelope. As the Golgi adopts a circumnuclear localization during muscle differentiation (see Section 4.2), future studies should aim to dissect the roles of the nuclear membrane and the Golgi for perinuclear ncMTOC function.

### 4.2. Golgi Organization and MTOC Formation are Coordinated Processes

The Golgi and the centrosome are spatially and functionally connected in most cells [151,193,194,195]. During cell polarization and cell migration, physical contact between the Golgi and the centrosome allows for a coordinated re-localization of both structures, e.g., towards the leading edge of the migrating cell. In addition, centrosome- and Golgi-derived microtubules cooperate with actin filaments to maintain Golgi structure [193]. Yet, the functional role of the connection between Golgi and centrosome is still poorly understood. In regards to polarization, it is believed that centrosomal microtubules facilitate polar/directional movement of Golgi-derived secretory vesicles, which for example, enables the recycling and directional positioning of receptors in the cell membrane.

Early studies regarding the MTOC status in striated muscle cells also examined the Golgi organization in these cells. The Golgi was found around the nuclei with its *cis*-side facing the nuclear membrane [173,196,197]. It colocalized with centrosomal material in these cells and electron microscopy suggested that the MTOC localizes between nuclear envelope and Golgi. Interestingly, the circumnuclear Golgi in myotubes is associated with stable detyrosinated microtubules [198] and resistant to nocodazole-induced microtubule depolymerization, a process known to disrupt the Golgi in mono-nucleated non-muscle cells [196,199], which argues for the importance of the microtubule network for Golgi integrity in myotubes. Pioneering live cell imaging studies of fluorescently labeled Golgi in differentiating muscle cells revealed that Golgi reorganization is a gradual process [200]. Delayed recovery after photobleaching indicated that the circumnuclear Golgi, which is often observed not as a continuous structure but as a pattern of short elements surrounding the nucleus [174,198], has lost the continuity inherent to the Golgi in proliferating cells [200]. Individual Golgi elements colocalized with ER exit sites, which mainly localize in a perinuclear belt in myotubes [200]. Interestingly, ER exit sites have been described to associate with growing microtubules in non-muscle cells [201]. Therefore, in muscle cells, accumulation of ER exit sites in a perinuclear pattern potentially contributes to the microtubule-dependent anchoring and stabilization of Golgi elements. The association between MTOC and Golgi reorganization during striated muscle differentiation requires more studying to determine, whether a hierarchy and/or interdependence between both processes exists, and to dissect the contributions of the nuclear envelope and the Golgi to microtubule nucleation (see Section 4.1). Notably, in 2013, it was shown that the presence of the Golgi at the nuclear envelope of cardiomyocytes is functionally important. Golgi membrane-bound phosphatidylinositol 4-phosphate is hydrolyzed by nuclear envelope-localized phospholipase Cε. The hydrolysis product diacylglycerol promotes activation of nuclear protein kinase D, which, in turn, stimulates hypertrophic pathways [202]. Yet, so far, no mechanism has been described how the Golgi is localized around the nucleus.

### 4.3. Centrosome Attenuation–Learning from Other Cells, Tissues, and Organisms

The formation of ncMTOCs is usually accompanied by centrosome attenuation. This includes the removal of PCM from centrioles and the reduction of centrosomal microtubule nucleation/anchoring but, in more extreme cases, also involves loss of centriole cohesion (cardiomyocytes [4]) or centriole degradation (spermatids and oocytes [203]). While mechanisms underlying the anchoring of MTOC components and microtubule nucleation at the nuclear envelope have been illuminated, it is unclear how the centrosomal MTOC is attenuated [2]. It appears possible that the processes that result in the establishment of the ncMTOC (e.g., recruitment of PCM proteins) are dominant over the mechanisms that maintain centrosomal MTOC activity. On the other hand, ncMTOC formation might require the loss of factors critical for MTOC function specifically at the centrosome, like it has been suggested for the loss of NEDD1 in keratinocytes [204]. In myoblasts, it has been reported that overexpression of nesprin-1α is sufficient to recruit the PACT domain and endogenous PCM1 [60,178]. However, recruitment of PCM1 was inefficient and other centrosomal proteins were not observed at the nuclear envelope. This is in agreement with the hypothesis that inactivation of the centrosomal MTOC is required to enable MTOC formation at the nuclear membrane. Yet, it is also possible that nesprin-1α is not sufficient to recruit MTOC components and that, for example, expression of cell type-specific adaptors and/or the alternative splicing of genes encoding centrosomal proteins (see next paragraph) are required to establish an ncMTOC.

In murine and human neurons, alternative splicing of ninein results in loss of the centrosome-targeting domain and, consequently, a non-centrosomal isoform of ninein. While non-centrosomal ninein promoted neural progenitor differentiation [205], its contribution to centrosome attenuation is not elucidated. In cardiomyocytes, PCNT is alternatively spliced, which results in a shorter isoform that preferentially localizes to the nuclear envelope but retains its centrosome-targeting (PACT) domain [4,206]. Moreover, ectopic expressed PACT domain localizes to the nuclear envelope in striated muscle cells, suggesting that loss of centrosome-targeting domains of PCM proteins does not underlie centrosome attenuation in muscle [60]. Yet, a systematic, comparative study of alternative splicing among centrosomal proteins in neurons and muscle cells would help to understand if splicing is involved in ncMTOC formation in these cell types.

One of the research fields that gives potential hints on how the centrosome might be attenuated in muscle cells is the study of PCM expansion and subsequent removal during mitosis [98]. A comparably small “basal” amount of PCM is present at centrosomes throughout the cell cycle and recent studies suggest that this ground state PCM is maintained as a highly ordered matrix limited by the extension of PCNT fibrils from the centriole into the cytoplasm (see Section 3) [98]. During mitosis, however, there is a rapid expansion of PCM volume and microtubule organization activity, which is induced by mitotic kinases (see Section 3). At the end of mitosis, the expanded PCM needs to be disassembled [115]. Matching the importance of phosphorylation for PCM expansion, a number of studies have implicated phosphatases in the disassembly process [207,208]. Recent studies in *C. elegans* identified protein phosphatase 2 (PP2A) to play a role in initiating PCM dissolution, which is then completed by rupturing microtubule-pulling forces exerted through cortically anchored dynein [207]. Interestingly, cortically anchored dynein is involved in nucleus positioning in myotubes (see 4.1). Although this anchoring is likely to be mediated by factors different from those in mitotic cells, it appears worthwhile to investigate if a similar loosen-and-rupture mechanism could be involved in centrosome attenuation in muscle cells. A study in *C. elegans* intestinal cells indicated that the loss of an active form (presumably phosphorylated by CDKs) of SPD-2 (CEP192 orthologue) from the centrosome allows ncMTOC formation at the apical membrane [209]. This further argues for posttranslational modifications of centrosomal proteins to play a general role in ncMTOC formation.

## 5. The ncMTOC in Muscle Function, Disease and Regeneration

Striated muscle cells encompass skeletal muscle myofibers and cardiomyocytes, which both assemble sarcomeres as their basic contractile unit. Sarcomeres consist of actin and myosin as well as a myriad of regulatory and scaffolding proteins and shortening of sarcomeres by an adenosine triphosphate (ATP)-dependent change in myosin conformation generates contractile force [210]. During skeletal muscle formation, progenitor cells commit to a myogenic fate and the resulting myoblasts then differentiate, exit the cell cycle, and form myotubes by iterative fusion [211,212]. Subsequently, multinucleated myotubes undergo maturation, which includes, among other processes, the pronounced regular assembly of sarcomeres, the formation of neuromuscular junctions (NMJs), and localization of myonuclei to the periphery [213].

In contrast to skeletal muscle, cardiomyocytes do not fuse but act as a functional syncytium due to extensive intercellular coupling. The heart is one of the first organs to develop and function, and fetal cardiomyocytes contribute to both processes by proliferation (heart growth) and contraction (heart function). During maturation, cardiomyocytes adopt an anisotropic, rod-like shape containing highly organized sarcomeres [214,215]. A hallmark of mammalian cardiomyocytes is the entry into a post-mitotic state after birth, in which cell cycle activity results in polyploidization and/or polynucleation. As a consequence, the majority of adult cardiomyocytes are tetraploid mononucleated in humans or tetraploid binucleated in rodents [216,217,218].

### 5.1. Nuclear Positioning in Skeletal Muscle 

When myotubes form by fusion, their nuclei first form a single cluster and subsequently spread and align, resulting in an even distribution along the myotube [219,220]. Subsequently, during maturation into myofibers, nuclei move to the periphery while maintaining maximal internuclear distance. An exception is the formation of multinuclear clusters at the NMJ and the myotendinous junction [221]. Muscle diseases often feature mislocalized nuclei [222] and perturbing myonuclear positioning impairs muscle function in model organisms [61,223,224,225]. Therefore, proper nuclear positioning appears to be critical for muscle function. Myonuclear movement has been recently reviewed in-depth in [219].

The importance of nuclear envelope proteins that connect the nucleus to the microtubule cytoskeleton for nuclear positioning has been underlined by a multitude of studies using a variety of cell types or model organisms [219,226] (Figure 3). The LINC complex has emerged as a key player in microtubule-based nuclear positioning [181,183]. In mammalian skeletal muscle, the KASH protein nesprin-1 has been first discovered to be critical for nuclear clustering at the NMJ, where its expression is higher compared to non-synaptic nuclei in mature muscle [227,228]. Subsequent studies expanded the role of KASH proteins to all aspects of myonuclear positioning [219,220,229]. Depletion of nesprin-1 impairs nuclear alignment in mammalian myotubes in vitro likely due to the loss of dynein and kinesin motor proteins from the nuclear envelope [178] which have been shown to be essential to regulate myonuclear movement [219]. Intriguingly, localization of motor proteins at the nuclear membrane also depends on PCM1, which itself requires nesprin-1 for localization to myonuclei [178] (Figure 2). This suggests that PCM1 acts as an adapter between microtubule-associated motors and the nuclear envelope. Interestingly, while PCM1 is conserved among vertebrate species, an obvious orthologue in *Drosophila* remains elusive. Consequently, it is unclear if PCM1-dependent mechanisms are conserved between mammalian and *Drosophila* muscle.

A conserved mechanism depending on the MAP ensconsin (Ens)/MAP7 and kinesin heavy chain (Khc)/Kif5b is especially important for myonuclear spreading [61]. In *Drosophila*, the ninein orthologue Bsg25D contributes to the function of the Ens-Khc complex [223]. While in mammalian cells PCM1 is a good candidate that could recruit the MAP7-Kif5b complex to the nuclear envelope, it remains elusive if an adaptor between Ens-Khc and KASH proteins at the outer nuclear membrane exists (Figure 2).

Microtubule-associated motors also contribute to myonuclear positioning by pulling forces transmitted by cortically anchored microtubules, which is collectively referred to as the cortical pulling mechanism [219] (Figure 3). Microtubules and dynein are anchored at the cell cortex via CLIP-190 and Raps/Pins [191,192]. Dynein movement along the microtubule results in cortex-directed pulling force on the nuclei. Microtubule anchoring at the nuclear envelope appears specifically important for this process, although a direct role of the nuclear envelope MTOC in the cortical pulling mechanisms has not been established so far.

Recent studies identified the interaction of short isoforms of nesprin-1, in particular nesprin-1α, with AKAP9 to be essential for non-centrosomal microtubule nucleation at the nuclear envelope (see Section 3.1) and showed that these microtubules are specifically required to spread nuclei along the length of the myotubes [60]. Sliding of antiparallel microtubules from neighboring nuclei has been suggested as the main role of nuclear envelope microtubules in myonuclear spreading [60,219] and it has been proposed that the MAP7-Kif5b complex contributes to this microtubule sliding [61,219]. A specific role for nuclear envelope microtubule nucleation in nuclear alignment, nuclear movement to the periphery or nuclear clustering at the NMJ has yet to be determined.

Considering the important role of the microtubule cytoskeleton for myonuclear positioning, surprisingly few human striated muscle diseases have been found to be caused by mutations in ncMTOC components so far [230]. The most prominent examples include mutations in *SYNE1* and *SYNE2*, encoding for nesprin-1 and nesprin-2, respectively, which result in Emery–Dreifuss muscular dystrophy (EDMD) [183,231]. While nuclear positioning is affected in skeletal muscle of EDMD patients, it remains unclear whether this is the main cause of muscle dysfunction in these patients. Mutations in nesprins might also disturb nuclear membrane architecture and chromatin organization, which results in aberrant transcriptional regulation. As recently discussed by Zhou et al. and Janin et al., a combination of defects in nuclear positioning as well as transcriptional and epigenetic regulation likely underlies muscle dysfunction in patients with *SYNE* mutations [183,231].

Mutations in the gene encoding for the GTPase Dynamin 2 (DNM2) cause a rare form of centronuclear myopathy (CNM), in which myonuclei are aberrantly located at the center of myofibers [232]. DNM2 is mainly known for its role in vesicle trafficking but has been suggested to be involved also in microtubule stabilization and centrosome function. While a role of DNM2 at the nuclear envelope has not been described up to now, it would be worthwhile to investigate if the microtubule network is affected in myofibers or cardiomyocytes from patients carrying *DNM2* mutations.

Mutations in other MTOC components (e.g., CEP215 and Dynein) have been associated with muscle dysfunction. However, a direct effect of these mutations on muscle physiology has not been described and the main pathological mechanism appears to be an effect on neuronal function, which, in turn, affects neuro-muscular coupling and control of muscle function.

### 5.2. Muscle Contractility and Mechanical Protection of Nuclei

Muscle contraction generates substantial force which exerts strain on cellular organelles [233]. Preserving intracellular architecture against stress from numerous contraction/relaxation cycles is pivotal for proper muscle cell function [233]. The nucleus, considered to be the largest and stiffest organelle in many mammalian cells [234], especially needs to be protected in order to maintain genomic integrity [235,236]. At the same time, forces that act on the nucleus are translated by the nucleoskeleton to modulate gene expression [183]. The microtubule cytoskeleton and the LINC complex play important roles in nucleus protection and in transmitting forces from the cell body/cytoplasm to the nucleus [183] (Figure 3). In *Drosophila*, the perinuclear microtubule network protects nuclei from deformation in a process that requires the microtubule plus-end tracker EB1 and the spectraplakin Shortstop (Shot) [237]. In part, this protection appears to rely on excluding sarcomeric actin from the direct vicinity of myonuclei. Furthermore, the LINC complex member MSP-300 (nesprin orthologue) forms a perinuclear “cage-like” structure which is stabilized by Shot [237]. It was observed that mechanical strain on muscle fibers deforms the perinuclear MSP-300 cage but does not affect gross nuclear morphology. In the absence of Shot, however, nuclei are deformed by strain which results in aberrant levels of regulatory nuclear factors such as lamins or heterochromatin protein 1. Furthermore, mutation of LINC complex members, such as Klar, dysregulates DNA endoreplication in myonuclei, which is stimulated by mechanical cues [238]. This leads to the aberrant expression of genes involved in contractile regulation. Interestingly, MSP-300 also functions in linking the peripheral myonuclei to the centrally located actomyosin network (i.e., the myofibrillar domain), which might have a synchronizing effect on muscle contraction and nuclear movement, thereby lowering strain stress on nuclei [225].

In cardiomyocytes, it has been shown that especially detyrosinated microtubules directly regulate muscle contractility by providing resistance to sarcomere shortening [239]. When detyrosination is inhibited, shortening and contractile velocity are increased. On the other hand, increased levels of detyrosinated microtubules, as well as intermediate filaments, can be detected in patients suffering from heart failure, and pharmacological inhibition of detyrosination results in a ~50% recovery of contractile function in failing cardiomyocytes [240]. How ncMTOC formation at the nuclear envelope contributes to the role of detyrosinated microtubules in cardiomyocytes has yet to be determined. A recent study showed that perinuclear microtubules exert compressive force on cardiomyocytes nuclei that is counteracted by desmin [62]. In the absence of desmin, microtubules form dynamic protrusions and drive nuclear involution that causes DNA damage, loss of association between nuclear lamina and chromatin, and ultimately large transcriptomic changes. The mechanism underlying the counteracting relationship between desmin and microtubules remains elusive.

Interestingly, a mutation in the gene for the MTOC component AKAP9 causes abnormalities in cardiac electrical conduction and contractility, referred to as Long QT syndrome [241]. While the pathological consequences of these mutations depend on the microtubule-independent role of AKAP9 to anchor protein kinase A [241], which, for example, phosphorylates ion channels, it would be of interest to examine if the microtubule network is affected in cardiomyocytes and skeletal muscle myofibers of Long QT patients.

### 5.3. Cell Cycle Exit and Regeneration

Establishment of the nuclear envelope MTOC in differentiating myoblasts and cardiomyocytes is associated with cell cycle exit. During skeletal muscle differentiation, myoblasts exit the cell cycle (i.e., they enter a postmitotic state) before fusing into myotubes, and these myoblasts exhibit a nuclear envelope MTOC [242]. However, a temporal order between cell cycle exit and ncMTOC formation in myoblasts has not been resolved so far. Cardiomyocytes establish a nuclear envelope MTOC shortly after birth and at the same time begin to enter a postmitotic state [243]. Yet, cardiomyocytes progress through one last cell cycle, which results in cytokinesis failure associated with aberrant mitotic microtubule distribution, mislocalization of RhoA and IQGAP3, as well as defective actomyosin ring anchorage and cleavage furrow ingression [244]. The cytokinesis failure leads to polyploidy and polynucleated cardiomyocytes [218]. It has been hypothesized that the last cardiomyocyte cell cycle occurs after the formation of the nuclear envelope MTOC [244]. Still, the putative causal relationship between cell cycle exit and nuclear envelope MTOC formation is unclear. Notably, while ncMTOC formation and cell cycle exit occur prior to contraction in skeletal muscle, fetal cardiomyocytes contract and at the same time exhibit a centrosomal MTOC and proliferate [4,245].

Under certain physiological, pathophysiological or experimental conditions, striated muscle cells can re-enter the cell cycle (G1/S phase), but generally fail to complete cell division [246,247,248]. Considering that cardiovascular diseases are among the leading causes of death worldwide, research in recent years was primarily focussed on inducing cardiomyocyte cell cycle re-entry to increase muscle mass [249]. Despite the importance of understanding the mechanisms underlying the post-mitotic state of mammalian cardiomyocytes for heart regenerative approaches, few studies have tried to unveil these mechanisms [245,250,251]. Zebrowski and colleagues reported that postnatal cardiomyocytes not only establish a nuclear envelope MTOC but also lose centrosome integrity resulting in centriole splitting [4]. Loss of centrosome integrity correlated with cardiomyocyte G0/G1 cell cycle arrest and experimental disruption of centrosome function by targeting PCNT resulted in a reduced proliferative potential of postnatal day 0 (P0)-isolated cardiomyocytes. Consequently, it was concluded that centrosome disassembly contributes to the cell cycle arrest in striated muscle cells. Furthermore, it was hypothesized that nuclear envelope MTOC formation is the cause of centrosome disassembly. However, so far it has been difficult to test this hypothesis experimentally as mechanistic knowledge of centrosome attenuation during ncMTOC formation is still sparse.

In the last 15 years, several studies have reported that postnatal cardiomyocytes can be induced to proliferate in vitro and in vivo [218,252,253]. While some of these results might be controversial, the data raise the question if ncMTOC formation and/or centrosome disassembly are reversible. Notably, yeast switch off ncMTOCs during mitosis to allow MTOC activity at their centrosome-like spindle pole bodies [254]. In addition, it has been shown that single centrioles can nucleate microtubules [255], and in mammalian female meiosis, oocytes form acentrosomal spindles [144,256]. Finally, it has been shown that mitosis can occur in mammalian cells in the absence of a centrosome [257,258,259,260,261]. However, daughter cells generated from centrosome-free bipolar spindles get arrested in the following G1 phase by a p53-dependent mechanism and fail to enter S phase [259,261]. In recent years, several centrosome-independent pathways for microtubule nucleation have been identified (see Section 3.2) that are mainly active during cell division. Activation of these pathways might be sufficient to support cell division in cells with a dominant ncMTOC and an attenuated/disassembled centrosome.

A few recent studies shed light on the fate of MTOCs and the role of microtubule organization during cardiomyocyte mitosis. It was found that cardiomyocyte mitosis resulting in binucleation (comparable to the last round of cell cycle in vivo; see above) is associated with diffuse patterns of EB1, γ-tubulin, and PCM1 at spindle poles in anaphase, which is in contrast to a distinct and focused pattern of these proteins in fetal proliferating cardiomyocytes [244]. Furthermore, analyses of EB1 staining in early anaphase suggested that significantly fewer astral microtubules reach the equatorial cortex in binucleating cardiomyocytes. Since astral microtubules, which specifically emanate from the centrosome, are fundamental for spindle orientation and thus proper cytokinesis [262], it was concluded that aberrant microtubule organization underlies cardiomyocyte cytokinesis failure. This conclusion is in accordance with the assumption that ncMTOC formation and centrosome disassembly in cardiomyocytes occur prior to a last cell cycle that results in polyploidization or polynucleation (see above). Another study observed that P3 cardiomyocytes experimentally stimulated to enter mitosis preferentially form multipolar spindles instead of normal bipolar spindles [4]. While this was attributed to single centrioles derived from centriole splitting that act as MTOCs, polyploid or polynucleated cardiomyocytes, that had duplicated their centrosomes in previous S phases, could not be excluded as a source for multipolar spindles. Indeed, a later study showed that binucleated cardiomyocytes induced to progress into mitosis form transient multipolar spindles which are then resolved into pseudo-bipolar spindles [263]. More importantly, this study confirmed that centrioles are located at the spindle poles of mitotic cardiomyocytes. Taken together, the aforementioned studies therefore strongly indicate that postnatal cardiomyocytes switch their dominant MTOC localization back to centrioles during mitosis and that ncMTOC formation at the nuclear envelope might be reversible. It would be of interest in future studies to determine if nuclear envelope breakdown at the onset of mitosis contributes to re-activation of centrioles as MTOCs, e.g., by releasing perinuclear centrosomal proteins. Notably, an earlier study by Srsen and colleagues showed that isolated myotube nuclei disassemble their nuclear envelope when incubated in mitotic *Xenopus laevis* egg extract, but centrosomal proteins remain localized to nuclear membrane patches [10]. This suggests that additional mitosis-related mechanisms promote MTOC activity at centrioles in cardiomyocytes.

Despite the re-activation of centrioles as dominant MTOCs during cardiomyocyte mitosis, multi- or pseudo-bipolar spindles are described to be error-prone, resulting in aneuploid progeny, due to merotelic attachments and lagging chromosomes [264]. Thus, inducing proliferation of postnatal cardiomyocytes, which have formed an ncMTOC and disassembled their centrosome, might result in aneuploid progeny. Therefore, it would be of importance to follow-up on the progeny of cardiomyocytes that divide with the help of re-activated centriolar MTOCs and to assess their viability. In addition, it needs to be examined whether the ncMTOC is properly re-established, as this might affect contractility of the new cardiomyocytes (see Section 5.2). Furthermore, Leone and colleagues observed that mitosis of binucleated cardiomyocytes results to a certain extent in the generation of daughter cells with exactly two centrioles [263]. It would be interesting to analyze in future studies if these daughter cells, which contain a potentially “correct” number of centrioles (i.e., four centrioles after S phase) to form a bipolar spindle, are less error prone in a subsequent mitosis and pose a potential target subpopulation for regenerative approaches.

## 6. Conclusions and Outlook

The regulation, function, and consequences of ncMTOC formation have gained increasing attention in recent years. Here, we summarized modes of microtubule nucleation at centrosomal and non-centrosomal sites as well as their functional outcome in striated muscle. While the centrosome is pivotal for animal cells to ensure fidelity and proper orientation of cell division, several microtubule-mediated processes are organized by centrosome-independent pathways. Furthermore, differentiated cells often exit the cell cycle, which eliminates the need for a centrosomal MTOC that safeguards cell division. Consequently, assigning dominant MTOC function from the centrosome to non-centrosomal sites in differentiated cells is evolutionary reasonable to meet new functional demands.

The nuclear envelope MTOC in striated muscle cells is remarkably dominant compared to the situation in other differentiated cells like neurons, which lack a defined ncMTOC. Several reasons appear possible why the unique structure and function of striated muscle requires the switch from a centrosomal MTOC to a nuclear ncMTOC: (1) Myonuclear positioning: While in mononuclear cells the centrosome contributes to nuclear positioning [265,266], moving multiple nuclei in the same cell is more complex. To spread nuclei along myotubes, microtubule-associated motors transport the nucleus in a cargo-like manner, and at the same time exert pushing and pulling forces on the nucleus via microtubules. In this context, centrosome protein-dependent motor recruitment, as well as microtubule organization, directly at the nuclear envelope allows timely force action from different directions and poses a clear advantage over centrosomal microtubule organization. Furthermore, it allows internuclear bidirectional force generation to push nuclei away from each other. (2) Contraction: A second unique feature of striated muscle is the large forces applied on the cell and its organelles by contraction. The nucleus in particular requires protection from mechanical stress to avoid DNA damage and disturbance of nuclear architecture, which potentially results in aberrant gene expression. While stabilizing modifications account for the major part of microtubule-mediated mechanical resistance, it appears logical that stable microtubules act most efficiently when they are organized in a circumnuclear fashion to form a protective cage around the nucleus and to divert intracellular mechanical stress in a symmetric fashion. (3) Mechanotransduction: The perception of mechanical cues and their translation into a specific cellular response, are particularly important for dynamic, force-generating tissues, such as cardiac and skeletal muscle. Microtubules are involved in transducing mechanical cues to the nucleus via the LINC complex. In response to these cues, gene expression is modulated. It could be hypothesized that dominant MTOC function at the nuclear membrane improves mechanotransduction and allows timely and more precise modulation of gene expression in muscle cells. (4) Intracellular organization: Sarcomeres, mitochondria, and t-tubules (invaginations of the muscle cell membrane) exhibit a highly organized architecture [59,267,268]. Microtubule organization from the comparably large nuclear surface might be advantageous for the proper regular organization of the muscle cell interior. (5) Hypertrophy: Circumnuclear Golgi is involved in the hypertrophic response of mature muscle cells to increased functional demands. While Golgi and MTOC re-organization are associated, a clear causal relationship in muscle has not been described so far. Considering that microtubule arrays help to position the Golgi in other cell types, it appears likely that ncMTOC formation at the nucleus enables the circumnuclear Golgi re-distribution.

Elucidating the formation and function of ncMTOCs, as well as the associated centrosome attenuation, might hold clinical potential: (1) Modulation of the ncMTOC might allow to influence muscle function during pathologies. Support for this hypothesis comes, for example, from the recent finding that targeting microtubule modifications can possibly prevent the loss of contractile function in failing human hearts [240]. (2) It is increasingly recognized that centrosome aberrations have an oncogenic role [269,270,271,272]. Canonically, centrosome abnormalities contribute to malignant transformation by disturbing the bipolar spindle and inducing aneuploidy [270]. Recently, however, it was shown that centrosome hyperactivity has aneuploidy-independent effects on migration, invasion, and metastasis [269,271,272]. Thus, while mechanisms of centrosome attenuation might not prevent mitotic centrosome (re-)activation (see Section 5.3), they potentially ameliorate consequences of centrosome aberrations in interphase cells. (3) Finally, understanding the mechanisms leading to centrosome attenuation and re-activation during mitosis in cardiomyocytes might allow to develop strategies to promote cardiac regeneration based on cardiomyocyte proliferation [218].

In summary, the field of ncMTOC research still harbors many unknowns. Several ncMTOCs have been discovered and their structure/composition has in part been elucidated. For several cell types, cellular functions have been identified that are associated or dependent on the ncMTOC. Yet, especially in striated muscle cells, many questions remain. (1) How does the ncMTOC specifically influence cellular functions, such as contractile resistance and mechanotransduction? Microtubules are pivotal for these processes, but the requirement for a nuclear envelope ncMTOC remains vague. (2) Does the ncMTOC directly influence microtubule modifications or does it generate specific microtubule subpopulations? So far, only AKAP9 has been shown to mediate microtubule nucleation at the nuclear envelope. At centrosomes and certain non-centrosomal sites, proteins like PCNT, CEP215, and ninein play important roles in microtubule organization. While these proteins are also recruited to the nuclear envelope, their function needs to be elucidated. (3) How and when is the centrosomal MTOC attenuated? Here it is important to note that centrosomal MTOC activity is also modulated during the cell cycle, and mechanisms of centrosome inactivation after mitosis are increasingly illuminated. As MTOC function at centrosome remnants in cardiomyocytes can be still activated by mitotic pathways, it would be logical to examine if centrosome attenuation in striated muscle involves similar mechanisms as centrosome inactivation post mitosis. (4) Does altered ncMTOC activity play a role in disease and can ncMTOC modulation be of therapeutic value?

Further elucidation of mechanisms underlying ncMTOC formation in striated muscle cells will provide adequate tools, such as mutants to specifically inhibit microtubule nucleation at the nuclear envelope or assignment of ncMTOC function to other cellular sites, to examine the specific role of a nuclear envelope ncMTOC in muscle function and pathology.

## Figures and Tables

**Figure 1 cells-09-01395-f001:**
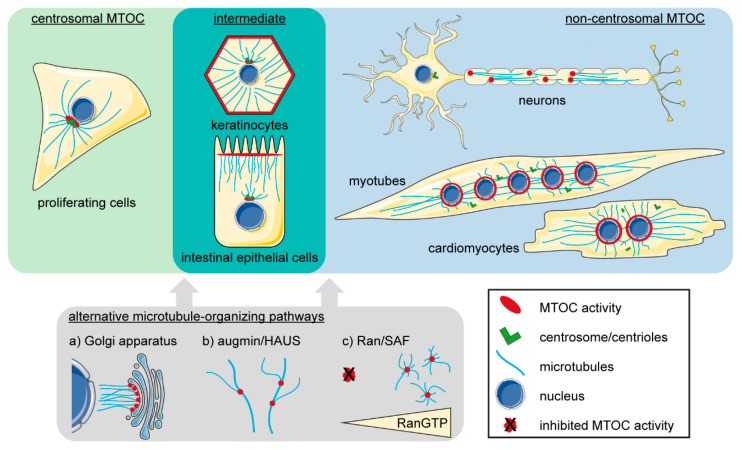
Examples of microtubule organization in animal cells (depictions are simplified; see main text for details). The centrosome is the dominant microtubule-organizing center (MTOC) in proliferating cells (green panel). Differentiated cell types like neurons or striated muscle cells exhibit various non-centrosomal MTOCs and attenuate/inactivate or, as described for cardiomyocytes, disassemble their centrosomes (blue panel). Certain types of epithelial cells are “intermediates”, i.e., they retain microtubule nucleation ability at centrosomes but anchor microtubules mainly at non-centrosomal sites (cyan panel). In addition, several alternative microtubule-organizing pathways (grey panel) are active in proliferating as well as differentiated cells (neurons in particular) that act in synergy with the dominant MTOC.

**Figure 2 cells-09-01395-f002:**
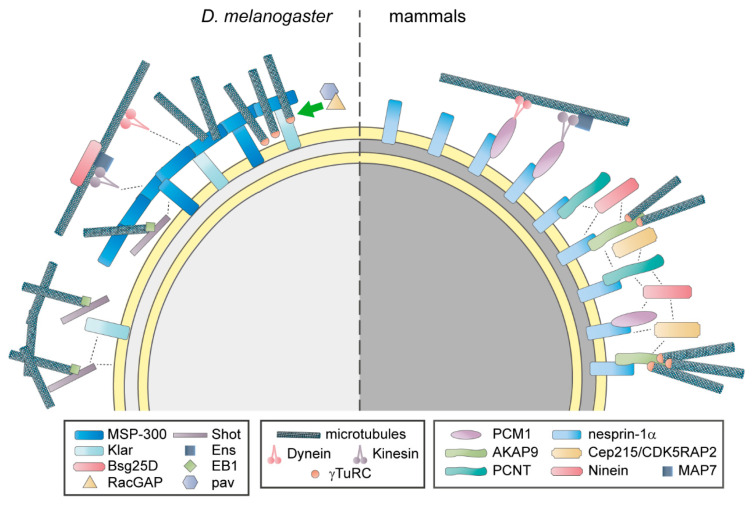
The nuclear envelope microtubule-organizing center (MTOC) in *Drosophila* (*D.*) *melanogaster* and in mammalian muscle cells. In *D. melanogaster*, the outer nuclear membrane proteins MSP-300 and Klarsicht (Klar), as well as a complex of the spectraplakin Shortstop (Shot) and the microtubule plus-end tracker EB1, cooperate to maintain a circumnuclear MSP-300 ring. Potentially, this ring is constituted by isoforms of MSP-300 that lack the nuclear envelope-targeting Klarsicht, ANC-1, and Syne homology (KASH) domain. The perinuclear microtubule network is maintained by the MSP-300 ring and the Shot-EB1 complex. Additionally, RacGAP and its binding partner Pavarotti (pav) promote perinuclear γTuRC localization. Dynein and a complex of kinesin, ensconsin (Ens), and Bsg25D mediate nuclear movement along microtubules. Klar and MSP-300 have been proposed as direct interaction partners of dynein and kinesin. In mammals, nesprin-1α anchors the centrosomal proteins pericentriolar material 1 (PCM1), pericentrin (PCNT), and A-kinase anchoring protein 9 (AKAP9) to the nuclear envelope. PCM1 connects the nucleus to microtubule-associated motors. AKAP9 is specifically required for microtubule nucleation, potentially by recruiting γ-tubulin ring complexes (γTuRC). Ninein and centrosomal protein of 215 kDa (CEP215/CDK5RAP2) localize to the nuclear envelope, but the underlying mechanisms are unclear. Ortho- and paralogues have the same shape and similar color. Dashed lines represent hypothetical interactions that underlie nuclear envelope localization of MTOC components. Please note that a large part of the *D. melanogaster* data has been obtained in myofibers in vivo, while mammalian data derives largely from in vitro experiments in myotubes. This possibly accounts for some of the differences in ncMTOC composition.

**Figure 3 cells-09-01395-f003:**
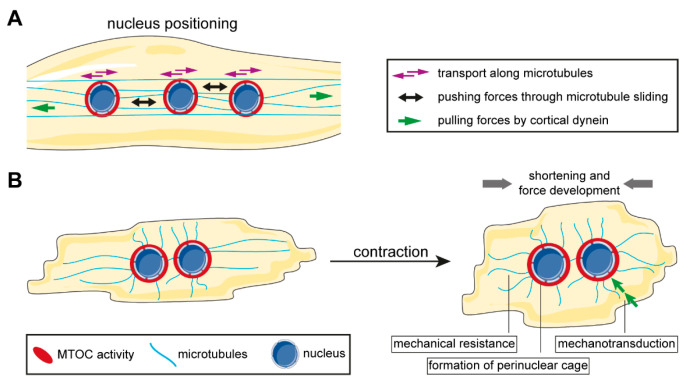
Contribution of the non-centrosomal microtubule-organizing center (ncMTOC) to striated muscle function (see main text for details). (**A**) The nuclear envelope ncMTOC promotes nuclear positioning in various ways. Microtubule-associated motors are recruited to the nuclear envelope to allow nuclear movement in a cargo-like manner along microtubules. Furthermore, perinuclear microtubules enable application of pushing forces through motor-mediated microtubule sliding as well as pulling forces exerted by cortically anchored dynein on the nuclei. (**B**) Microtubules fulfill several functions during muscle contraction. Firstly, stable microtubules modulate contractility by providing mechanical resistance. Secondly, a perinuclear cage consisting of microtubules and MTOC proteins protects nuclear integrity. Thirdly, microtubules link the extracellular space and the contractile apparatus to the nucleus via the linker of nucleoskeleton and cytoskeleton (LINC) complex, which allows transmission of mechanical cues to regulate gene expression.

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
