# Peer review of "Microtubule Organization in Striated Muscle Cells"

_cells, 2020, doi:10.3390/cells9061395_

Round 1

Reviewer 1 Report

The review by Becker and colleagues focuses on how striated muscle cells utilize centrosome-independent mechanisms to produce a specific distribution of microtubules and nuclei.  These mechanisms involve the use of the nuclear membrane as a non-centrosomal microtubule organizing center (ncMTOC) and various alternative microtubule organizing pathways.   The review is well written, informative, and comprehensive.  While suitable for publication, I do have a number of minor comments that should be addressed.

1.Line 95.  The authors comment on how different tubulin isoforms impart different properties on microtubules.  They might also want to cite the work from the Roll-Mecak lab (Vemu A, Atherton J, Spector JO, Moores CA, Roll-Mecak A. Mol Biol Cell. 2017 Dec 1;28(25):3564-3572.).

2.Line 111.  It would be more accurate to say that chromosomes segregate (rather than divide).

  1. Line 148. The authors talk about how dynein localized at the cell cortex exerts forces on astral microtubules to shape the spindle. It is my understanding that these forces are involved in positioning the spindle.  Isn’t this correct?

4.Figure 1.  Some aspects of this figure are not explained in the legend.  For instance, what is the X over the red dots in the panel labeled Ran/NuMA supposed to represent? Do the different panel colors have any meaning?

  1. Line 219. The authors use the term “microtubule spindle formation”. I assume they meant to say “mitotic spindle formation”.

  1. Line 301.Typo.  The phrase “to date” appears twice in the same sentence.

  1. Line 333.The authors mention that Pavarotti, a kinesin, plays a role in perinuclear localization of gamma-tubulin but earlier (line 306) mention that microtubules are not required to form a perinuclear MTOC.  These observations seem at odds.  Could the authors comment?

  1. Figure 2. MSP-300 is drawn as both integral nuclear envelope protein and a cytosolic protein. Is this true?  If so, how is this possible (alternative spliced isoforms?).

  1. Line 371. The authors state that a physical connection between the Golgi and centrosome is required during cell polarization and migration but do not state what this connection is specifically required for.  

  1. Line 406. Typo. functionally important

  1. Line 406.The statement “is a critical process of cardiac hypertropy” is vague.  I’m not sure what is meant by this.  Could the authors re-write this sentence to make their thoughts clearer?

  1. Line 421. The authors cite work showing that over-expression of nesprin is unable to efficiently recruit PCM1 and other centrosomal proteins to the nuclear envelope. They interpret this as evidence that inactivation of the centrosome also has to occur in order to form the nuclear MTOC.  I don’t see this as evidence of a requirement for centrosome inactivation.  All this experiment says is that over-expression of nesprin by itself is not sufficient to form the nuclear ncMTOC.  It might be that other proteins are also required for recruitment of MTOC proteins to the nucleus, or perhaps that different isoforms of MTOC components need to be expressed in these cells.  This should be re-written to reflect that.

  1. Line 578. The authors state that oocytes duplicate. Oocytes do not duplicate but they do undergo meiotic cell division on an acentriole spindle and extrude polar bodies.  I believe this is what the authors meant to say.  The sentence should be reworded.

  1. Line 610.What are C2C12 cells?

Reviewer 2 Report

This is an interesting, informative, and well-written review on the ncMTOC in striated muscle cells that I think will be useful to those interested in ncMTOCs generally, and in the muscle nuclear surface ncMTOC specifically. The three figures are well-constructed and organized and aid in the communication of ideas presented. Particularly useful and informative are the ideas expressed for why the ncMTOC’s role in positioning nuclei in myotubes and cardiomyocytes is physiologically important.

Some points to improve the review:

  1. The sections that preface the ultimate discussion of myotube and cardiomyocyte ncMTOC structures and functions (sections 2.1 and 2.2) are a bit expansive and could be trimmed to remove information that does not get connected to the ncMTOCs that are the focus of the review. For section 2, the points that MTs are important, dynamic, and regulated by a host of proteins and PTMs can be abbreviated. Only include the factors that have relevance to the muscle ncMTOC and that return to this context in the later sections. For example, discussion of the depolymerizing kinesins (p. 4) is not necessary since it is not integrated into the muscle ncMTOCs.

  1. Figure 1: I don’t think the neuron is depicted accurately. The ncMTOCs are drawn in the axon, but I believe that the ‘outposts’ or ‘golgi outposts’ in neurons are located in the dendrites. Axonal MTs are typically stabilized and oriented with the plus-ends away from the soma. In addition, neurons have a centrosome that is inactive (perhaps similar to the keratinocytes and intestinal epithelial cells depicted here).

  1. For the discussion of NEDD1 (aka GCP-WD) as a g-TuRC adaptor regulator, authors should also cite Luders et al 2006 (line 203).

  1. In the section that discusses golgi ncMTOCs (section 3.3), I recommend also citing contributions from Kaverina and Akhmanova labs; perhaps add their reviews on the subject (Zhu and Kaverina, 2013; Sanders and Kaverina, 2015; Wu and Akhmanova, 2017).

  1. In the section on epithelial cell ncMTOCs (starting on line 250) recommend adding this review: Toya and Takeichi, Organization of non-centrosomal microtubules in epithelial cells. Cell Struct Funct. 2016.

  1. Regarding the molecular structure and MT assembly functions of the muscle ncMTOC, the authors should elaborate more about the functions or requirements of the components. For example, AKAP450 is essential as the authors point out, and is anchored by Nesprin-1alpha, but it appears that PCM proteins that it is required to anchor, like Cep215 and Pericentrin, are not essential for MT assembly (Gimpel et al 2017).

  1. Related to the previous point, Figure 2 depicts the g-TuRC associated with AKAP9, but has this been shown? AKAP9 appears to anchor Cep215 and Pericentrin, and perhaps g-TuRC is recruited through those proteins. Also, I could not find in the literature where g-tubulin was shown to be important for the muscle ncMTOC. A recent paper reported a nuclear membrane ncMTOC in Drosophila fat body cells (Zheng et al 2020), and they showed that g-tubulin was not required, but instead other MT regulators are. Perhaps some discussion/speculation about what regulates MT assembly at the muscle could be added in connection with this idea.

  1. Additional points on Figure 2: Msp300 and Klar were shown to associate and cooperate, which could be depicted for the Drosophila MTOC (Ehanany-Tamir et al 2012). There has been a lot of focus on PCM1 in mammalian muscle, but its role in Drosophila has not been addressed (maybe can be incorporated into Figure 2 or mentioned in section 5.1).

  1. The paragraph that begins on line 379 is very long and ultimately does not seem to tie together the golgi association with the nucleus with the MTOC – what is the connection? Please tie this paragraph into the flow better, clarify what the point is, or just remove it.

  1. At the beginning of Section 5, readers unfamiliar with myotube and cardiomyocyte development would benefit from a brief introduction to muscle development and cell morphology.

  1. The heading for section 5 includes disease, but there is little discussion of it. A useful addition would be a section that connects muscle ncMTOC function to disease. For example, mutations in SYNE1, which encodes Nesprin-1, causes congenital muscular dystrophy (EDMD4; OMIM 612998), and Long QT Syndrome 1 is linked to mutations in AKAP9/AKAP450 (OMIM 192500). A summary of these and other disease connections to the known components of the ncMTOC would be very informative and important to include.

  1. Section 5.2 should incorporate, or at least cite, this paper from the Volk lab: Wang, Stoops, …, Volk. Mechanotransduction via the LINC Complex Regulates DNA Replication in Myonuclei. J Cell Biol.

  1. The discussion of centrosome dysfunction and its links to aneuploidy, cancer, metastasis, etc, that starts on line 674 needs citations.

  1. These additional papers should be cited:
  • Zhang, Felder, …, Ju Chen. Nesprin 1 is critical for nuclear positioning and anchorage. Hum Mol Genet. 2010.
  • Holt, Fuller, …, Morris. Nesprin-1-alpha2 associates with kinesin at myotube outer nuclear membranes but is restricted to neuromuscular junction nuclei in adult muscle. Sci Reports. 2019.
  • Tassin, Maro, and Bornens. Fate of Microtubule-organizing Centers during Myogenesis In Vitro. J of Cell Biol.

Additional minor corrections by line:

93 – in vicinity => in thevicinity

104 – add “to” after “due”

157 – similar => similarly

160 – family => families

170 – add “in” after “described”

281 – declined => declines (or change the tense of the rest of the sentence)

299 – if => whether

303 – delete “to date” (repeated from beginning of sentence)

310 – Nesprin-1 is introduced without mention of the LINC complex, which is introduced on line 466. Perhaps rearrange to introduce LINC earlier?

360 – via the cytoplasmic => via cytoplasmic

414 – involved also => also involve

429 – in shorter => in a shorter

436 – might attenuated => might be attenuated

478 – Start a new paragraph between “envelope. A” since you’re no longer talking about nuclear membrane proteins.

487 – direct => directed

502-503– Especially the nucleus, … needs to be protected => The nucleus, … especially needs to be protected

531 – delete “especially”

539 – microtubule => microtubules

568-569– had been => was

572 – centrosomes => centrosome

600-601– remove commas after “cardiomyocytes” and “phases”

618 – follow-up the progeny => follow-up on the progeny

620 – if => whether

648 – are => is

649 – Especially the nucleus requires => The nucleus especially requires

652 – efficient => efficiently

667 – it appears likely possibility => it appears a likely possibility (or it appears likely that)

675 – spindle an inducing => spindle and inducing

679 – mechanism => mechanisms
